# 2D honeycomb transformation into dodecagonal quasicrystals driven by electrostatic forces

Sebastian Schenk[1], Oliver Krahn[1], Eric Cockayne [2], Holger L. Meyerheim[3], Marc de Boissieu[4], Stefan Förster [1] ✉ & Wolf Widdra [1]

Dodecagonal oxide quasicrystals are well established as examples of long-range aperiodic order in two dimensions. However, despite investigations by scanning tunneling microscopy (STM), low-energy electron diffraction (LEED), low-energy electron microscopy (LEEM), photoemission spectroscopy as well as density functional theory (DFT), their structure is still controversial. Furthermore, the principles that guide the formation of quasicrystals (QCs) in oxides are elusive since the principles that are known to drive metallic QCs are expected to fail for oxides. Here we demonstrate the solution of the oxide QC structure by synchrotron-radiation based surface x-ray diffraction (SXRD) refinement of its largest-known approximant. The oxide QC formation is forced by large alkaline earth metal atoms and the reduction of their mutual electrostatic repulsion. It drives the $n = 6$ structure of the 2D $Ti_2O_3$ honeycomb arrangement via Stone–Wales transformations into an ordered structure with empty $n = 4$, singly occupied $n = 7$ and doubly occupied $n = 10$ rings, as supported by DFT.

Dodecagonal oxide quasicrystals (QCs) are two-dimensional metal-oxide ultrathin films that exhibit a sharp 12-fold diffraction pattern[1,2]. They have been identified for Ba-Ti-O and for Sr-Ti-O metal-oxide layers on hexagonally closed-packed metal substrates[1–8]. The dodecagonal symmetry is associated with a distinct square-triangle-rhomb tiling that was derived independently as a mathematical model a quarter of a century ago by Niizeki and Mitani and by Gähler, the Niizeki-Gähler tiling (NGT)[9,10]. The bright spots observed by atomically resolved STM images for the Ba-Ti-O and the Sr-Ti-O QCs closely match the vertices of this mathematical model[11]. As in many other QC systems, oxide QCs come along with a family of approximants that are periodic arrangements composed of the very same building blocks. They differ in complexity and unit cell size, classified by the number of tiling elements. The known approximants range from 4:2:0 to 48:18:6 with respect to the number of triangles, squares, and rhombi within the unit cell[2–4,12–15]. How these approximants are related to or evolve from

oxide quasicrystals are open questions, since the atomic structure is still under debate. Earlier studies on the 4:2:0 approximant, the smallest square-triangle tiling commonly known as sigma phase, suggested that Ti atoms decorate the triangle and square vertices for Ba-Ti-O on Pt(111) based on STM, SXRD and density functional theory (DFT) calculations[4,5]. In contrast, a DFT study proposed a structural model for various approximants where the tiling vertices are decorated by Ba instead[3], for which an uplifting into the four-dimensional hyperspace has been proposed also[16]. A third structure with a different stoichiometry was proposed by Yuhara et al. based on STM, XPS and Rutherford backscattering experiments[6,7]. This controversial situation arises from the lack of experimental data for a full structure determination. Though atomically resolved STM images pin down the dodecagonal quasicrystalline vertex structure as seen by bright features of protruding atoms, their chemical nature is debated. Furthermore, despite several attempts, information about the second and third atomic

[1]Institute of Physics, Martin-Luther-Universität Halle-Wittenberg, 06099 Halle, Germany. [2]Material Measurement Laboratory, National Institute of Standards and Technology, Gaithersburg, MD 20899, USA. [3]Max Planck Institute of Microstructure Physics, 06120 Halle, Germany. [4]Universite Grenoble Alpes, CNRS, SIMaP, St Martin d'Heres, France. ✉e-mail: stefan.foerster@physik.uni-halle.de

species in the ternary oxide is not experimentally accessible either from STM or from non-contact AFM[1,17].

In the present work using SXRD, we will resolve these issues and present a complete structural analysis in accordance with DFT for a large-scale quasicrystal approximant. From the proposed structure we derive a tiling decoration scheme that also applies to the quasicrystalline NGT. For the formation process of the dodecagonal oxide QC, we identify Stone–Wales transformations as key elements that drive 2D hexagonal honeycomb lattices into QC approximant structures with square-triangle tiling. The Stone–Wales defect is known as a low-energy structural transformation in many 2D honeycomb lattices, including graphene, h-BN, silicene, and $Ti_2O_3$[18,19]. It converts four adjacent six-fold honeycomb rings, $Ti_nO_n$ with $n = 6$, into two $n = 7$ and two $n = 5$ rings. For many 2D materials such as, e.g., graphene, the transformation stops upon a few Stone–Wales conversions and results in a disordered structure consisting of $n = 5$, 6, and 7 rings. Fully random, but also continuous structures with a broad ring size distribution are known from 2D silica films, which are 2D analogs of the Zachariasen model for glass formation[20,21]. For the $Ti_2O_3$-based QC and its approximants we demonstrate that their long-range order results from the self-organization of alkaline earth metal atoms stabilizing $Ti_nO_n$ ring sizes with $n \geq 7$.

## Results

### Structure determination for a large-scale approximant

For the dodecagonal QC structure determination, we focus on the largest-known oxide QC approximant[12]. It shows a periodic tiling with 48 triangles, 18 squares, and 6 rhombi in the unit cell (Fig. 1a, c, e). It is prepared by deposition of Sr and Ti on Pt(111) followed by post-annealing in an oxygen atmosphere of $10^{-4}$ Pa. The long-range order forms upon high-temperature annealing at 1250 K under ultrahigh vacuum conditions (see Supplementary Fig. 1). Atomically resolved STM images clearly show the vertex structure

of the square-triangle-rhomb tiling as depicted in Fig. 1a. The Fourier-transform (FT) taken from large-area STM data (Supplementary Fig. 2) identifies the large unit cell of an almost square lattice as fine grid with reciprocal lattice vectors of 0.14 Å$^{-1}$. Twelve reflections of higher intensity are located at a distance of 1.0 Å$^{-1}$ from the center. They resemble the characteristic diffraction intensities of the dodecagonal QC. The close resemblance of the approximant and the QC diffraction pattern indicates their structural similarity where 48 vertex atoms are arranged in 72 tiles with only minor modifications from the dodecagonal NGT (Fig. 1c). The latter include chains of three rhombi in the approximant, while pairs of rhombi dominate the oxide QC tiling meeting in the center of a characteristic dodecagon[11]. Our SXRD experiments carried out at the synchrotron radiation facility Soleil allow to determine the unit cell dimensions more accurately (Supplementary Fig. 3): the periodic square lattice as shown in Fig. 1b has lattice parameters of $a = 44.3$ and $b = 43.2$ Å equivalent to a commensurate $\begin{pmatrix} 16 & 0 \\ 9 & 18 \end{pmatrix}$ superstructure with respect to the primitive hexagonal unit cell of the Pt(111) substrate. The structure with this tiling pattern has p2gg plane-group symmetry due to the presence of two orthogonal glide lines. Therefore, the size of the symmetrically independent part is equal to one-quarter of the unit cell only (filled region in Fig. 1a). Owing to the fact that the rectangular p2gg unit cell grows on the p6mm symmetric surface, six rotational and mirror domains of this superstructure exist. From the full SXRD reciprocal space maps, which are partly shown in Fig. 1e (Supplementary Fig. 3 for full data), the integrated intensities of 460 reflections of type (hk0) were collected which reduce to 182 symmetry-inequivalent ones with an average agreement factor of 0.15. After correction for instrumental factors, the structure factor magnitudes $|F_{obs}(hk0)|$ were considered for the structural analysis. For the structural

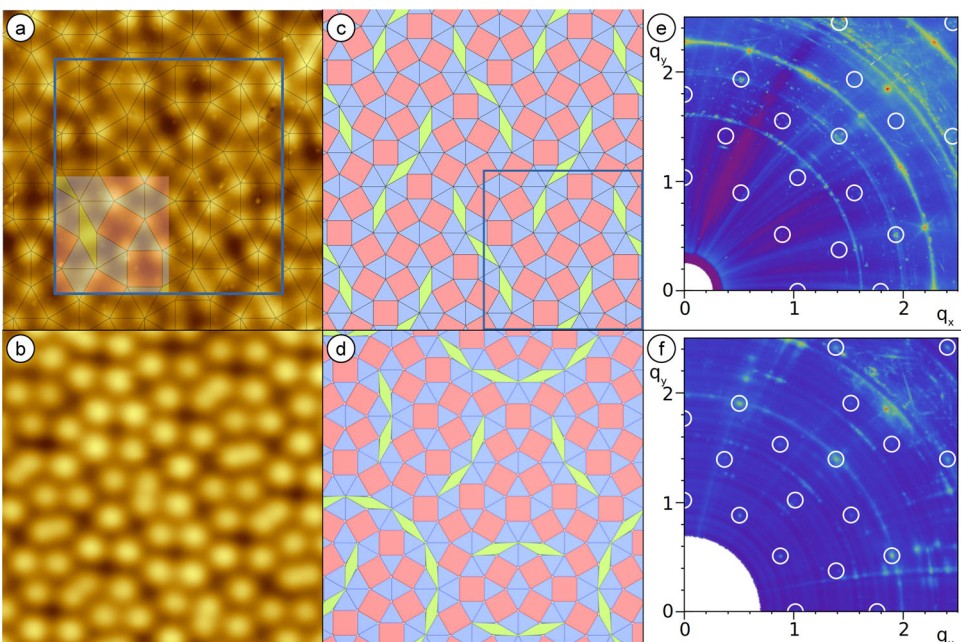

**Fig. 1 | Comparison of the 48:18:6 approximant structure with the oxide quasicrystal. a** The atomic-resolution STM image of the Sr-Ti-O approximant reveals 48 protrusions in the unit cell (blue square). The semi-transparent tiling highlights the asymmetric unit. **b** The vertex atoms of the Ba-Ti-O oxide quasicrystal arrange in similar elementary tiles. **c** The approximant tiling is characterized by chains of three rhombuses, while pairs of rhombuses meeting in the center of a dodecagon dominate the QC tiling (**d**). In reciprocal space, the most intense spots of the

approximant (**e**) are centered around the spot positions of the dodecagonal pattern (**f**) as marked by white circles in the SXRD data. The close correspondence of spot positions and intensities for both structures is demonstrated in Supplementary Fig. 8. The spot splitting in (**e**) results from the occurrence of multiple domains of the approximant structure. **a** 6 × 6 nm², 0.1 nA, 1.0 V. **b** 6 × 6 nm², 0.015 nA, −1.0 V.

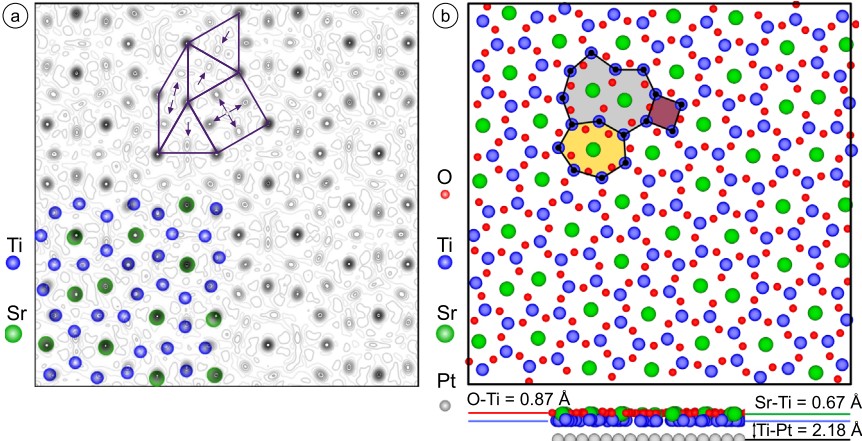

**Fig. 2 | Atomic structure of the 48:18:6 approximant. a** SXRD-derived electron density map calculated from the best fit model for the unit cell of the 48:18:6 approximant in Sr-Ti-O on Pt(111). The heavy Sr atoms (green) reside at positions of the highest electron densities. By connecting the Sr atoms the tiling seen in STM is formed (emphasized in the upper part). The weaker electron densities are attributed to Ti atoms (blue). Their positions inside the tiling elements are marked by purple arrows. **b** Relaxed atomic structure containing the oxygen sublattice calculated by DFT in top and side views. The Ti atoms are attracted toward the Pt interface. The average height of the Sr and O atoms above the Pt substrate is by 0.67 and 0.82 Å larger as compared to Ti. The characteristic features of this structure are $Ti_nO_n$ rings with $n = 4$, 7 and 10 hosting 0, 1 and 2 Sr atoms, respectively (highlighted in purple, orange and gray).

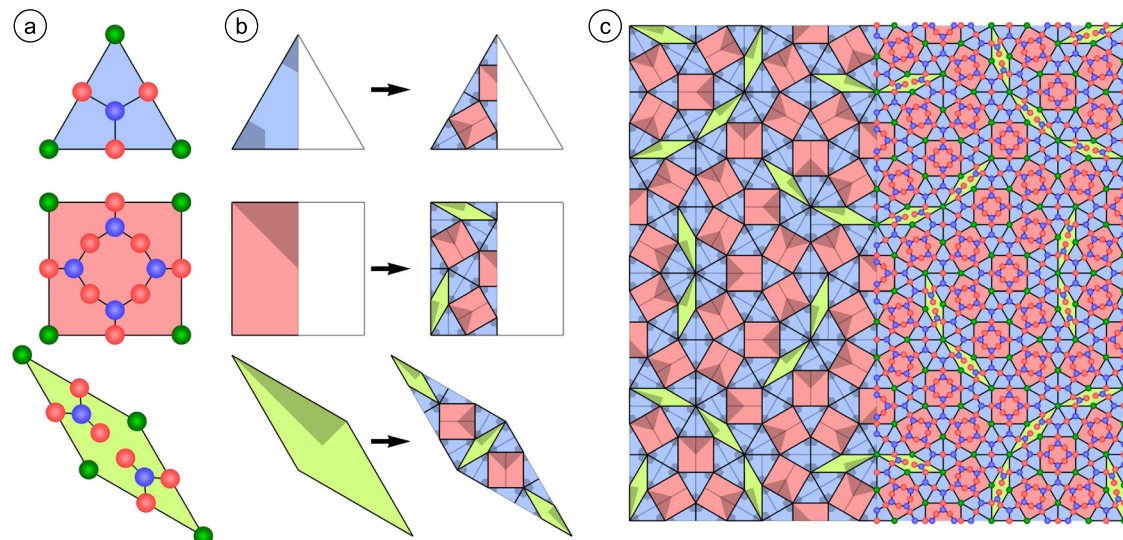

**Fig. 3 | Generalization of the atomic structure to all related ternary oxides. a** Decoration scheme for the three tiles of the oxide QC. Green, blue and red spheres represent A-side (Sr or Ba), Ti, and O atoms, respectively. **b** Inflation rule for the Niizeki-Gähler tiling of oxide quasicrystals based on three tiling elements. Shaded areas indicate the symmetry inherent to the tiling elements. The definition of the recursion rule based on half a triangle and half a square determines an overlap-free dodecagonal square-triangle-rhomb tiling. **c** Square supertile after three iterations of recursion. The decoration scheme of (**a**) has been applied to the right part of (**c**).

refinement process, two complementary models have been used as a starting point, where the vertex positions of the tiling are occupied either by the alkaline earth metal Sr or by Ti atoms. These complementary assignments correspond to the proposed oxide QC models based on DFT or STM, respectively[3,4]. Despite the complexity of the large unit cell with 72 tiling elements for an SXRD structural refinement, the positions of the vertex atoms as well as those of the second (non-vertex) atomic species have been optimized by a constrained least-squares refinement using the program SHELXL[22]. Oxygen atoms within the structure are not considered due to their small scattering cross-section. With regard to all agreement factors (Goodness of fit (GOF), weighted residuum (wR2), and unweighted residuum (R1))[22,23], the structural model including Sr atoms occupying the vertex positions

leads to substantially better fits (GOF = 1.5, wR2 = 0.21, R1 = 0.11) as compared to the alternative with Ti atoms at the vertices (GOF = 4.2, wR2 = 0.73, R1 = 0.45)[4,5].

This leads us to conclude that the SXRD analysis is in agreement with the theoretical model developed by Cockayne et al.[3] The arrangement of the Sr and Ti atoms can be directly visualized by the calculation of the charge density contour map, $\rho(x, y)$, using the observed structure factor magnitudes ($|F_{obs}|$) and the calculated scattering phases $\alpha$ derived from the structural model (due to the centrosymmetry of the z-projected structure, the condition $\alpha = 0$ or $\pi$ holds). The highest-density features as marked in green in Fig. 2a correspond to the positions of the heaviest atom, Sr. They form the vertices of the square-triangle-rhomb tiling. The Ti atoms are located at positions of the second highest electron density, as emphasized in

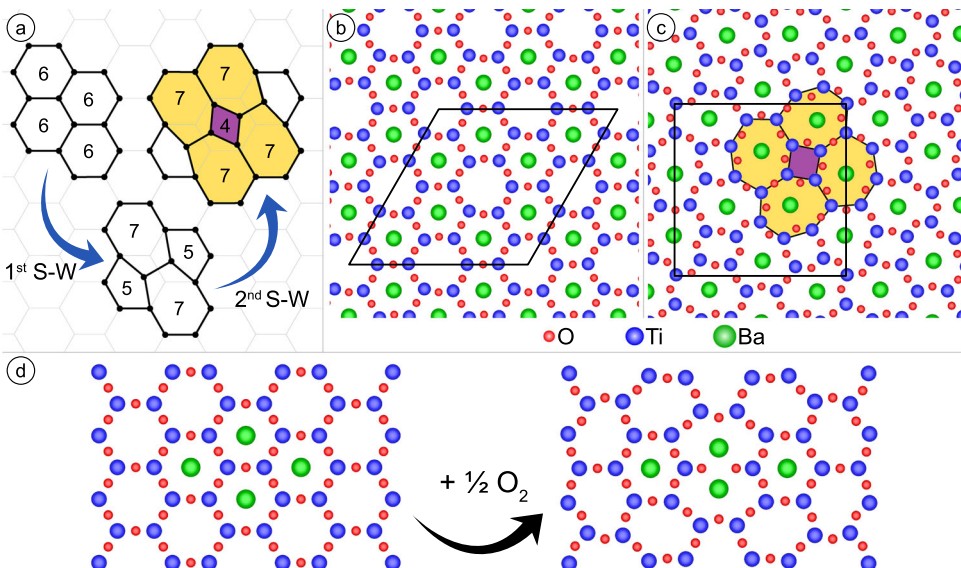

**Fig. 4 | Stone–Wales transformation converting honeycomb structure into square-triangle tiling and rhombus formation by incorporation of additional oxygen. a** Transition from a honeycomb lattice by subsequent Stone–Wales (S–W) transformations into a structure of four- and seven-member rings. **b** Comparison of the DFT relaxed structures of the Ba-decorated $Ti_2O_3$ honeycomb structure and **c** the Ba-Ti-O sigma-phase approximant on Pt(111). The DFT calculations were performed for these commensurate $Ba_8Ti_{24}O_{36}$ supercells ontop of three layers of Pt(111) (Pt not shown). The lowest energy phase is the sigma-phase approximant. **d** The addition of one oxygen atom to a Ti-O-Ti edge of two adjoining hexagons leads upon relaxation to the $n = 10$ ring of the oxide QC (compare central $n = 10$ ring with those in Fig. 2).

blue and by the black arrows in Fig. 2a. Note that the weak intensities in between the refined atomic positions as well as the small deviation between ideal positions and maxima of the calculated electron density result from Fourier series truncation errors due to the limited number of experimentally available diffraction intensities.

## Theoretical description
DFT calculations for the 48:18:6 approximant on Pt(111) provide further insights into the full atomic structure especially the oxygen positions. The fully relaxed structure is shown in Fig. 2b and can be understood as a partially Sr-decorated $Ti_2O_3$ network, in which every Ti atom is coordinated by three O atoms at an average distance of 1.89 Å (standard deviation 0.02 Å). This value is slightly larger than the 1.84 Å reported for DFT calculations of freestanding $Ti_2O_3$ in a planar honeycomb geometry[24]. The positions of each species above the Pt surface are quite uniform: Sr 2.85 Å (standard deviation 0.02 Å), Ti 2.18 Å (0.05 Å), and O 3.00 Å (0.06 Å). The average Sr-Sr distance corresponding to a tile edge is 6.75 Å (0.18 Å), which is in line with the experimentally determined value of 6.72 Å for the oxide QC[2]. The average Sr-Sr distance corresponding to the short diagonal of the 30° tile is 3.79 Å (0.07 Å), about 8% larger than that for a perfect 30° rhombus of edge length 6.75 Å but comparable to the 3.905 Å lattice parameter of perovskite $SrTiO_3$. Merchan et al. also reported results for a DFT relaxation of the Cockayne et al. structure model, and found similar structural features in spite of differences in the computational details[12].

Whereas $Ti_2O_3$ honeycomb structures consist of hexagonal arrangements of planar $Ti_6O_6$ rings, the oxide QC approximant contains $Ti_nO_n$ rings with $n = 4$, 7 and 10 only (colored in Fig. 2b). The smallest $n = 4$ rings are empty, whereas the larger $Ti_7O_7$ rings are decorated and stabilized by Sr, as we show below. The $n = 10$ pores host two Sr and two additional O atoms. These additional oxygen atoms screen the positively charged Sr cations thus allowing a shorter Sr-Sr next-neighbor distance in these rings.

From the structure solution presented, a decoration scheme for all tiles of the two-dimensional ternary oxide phase is confirmed, which universally applies to the quasicrystalline NGT and all related approximant structures (Fig. 3a)[3]. By combining the tiling element ratio in the respective unit cell with the atom density given by the decoration scheme, the stoichiometry of a given structure can be derived. For the 48:18:6 approximant a composition of $Sr_{48}Ti_{132}O_{204}$ and Sr:Ti:O ratio of approximately 0.364: 1: 1.545 is obtained. For the oxide QC, the Sr:Ti:O ratio is very similar: $(\sqrt{3} - 1)/2{:}1{:}(3\sqrt{3} + 1)/4 \approx 0.366{:}1{:}1.549$.

The atomic network of the oxide QC can be constructed from a combination of recursion and decoration (Fig. 3b, c). The definition of the NGT based on a recursion rule using half a triangle, half a square and the full rhombus given in Fig. 3b is the first that defines an overlap-free NGT, in contrast to earlier attempts[25]. The tiling as well as the atomic network formed from decoration are illustrated in Fig. 3c after three inflation steps. The equivalent tiling decoration scheme of the 48:18:6 approximant and the oxide QC leads to a high level of agreement of the corresponding diffraction intensities of both systems as demonstrated in Supplementary Fig. 8. While the average tiling decoration is well-defined from the DFT results, there are some local deviations. In particular, the three oxygen atoms surrounding a given Ti are frequently rotated clockwise or counterclockwise with respect to their average positions, associated with a tendency for each Sr atom to form bonds of length 2.4–2.7 Å with 3 or 4 of its oxygen neighbors.

## Discussion
### Driving force toward quasicrystal approximant formation
Many 2D threefold coordinated networks display disorder that can be described as the application of one or more Stone–Wales-type transformations to a hexagonal honeycomb lattice[19,26–28]. The Stone–Wales defect, discussed originally for icosahedral $C_{60}$, twists one edge in the network and converts four adjacent $n = 6$ honeycomb rings into $n = 7$ and $n = 5$ rings[18]. Typical threefold networks with disorder are characterized by an approximate Gaussian size distribution peaking at $n = 6$. In sharp contrast to these previously reported structures, the oxide QC approximant structure solved here, is composed of $Ti_nO_n$ rings characterized by $n = 4$, 7, and 10 only (hosting 0, 1, and 2 alkaline earth metal atoms respectively). The stabilization of such low symmetry $Ti_nO_n$ rings with $n = 7$ and $n = 10$ by the alkaline earth metal

atoms Sr or Ba, which will be called A-type atoms, are the essential features to explain.

Ultrathin film $Ti_2O_3$ has the well-known hexagonal honeycomb structure with $n = 6$ rings whereas the sigma 4:2:0 approximant (under the decoration scheme in this paper) has only $n = 4$ and $n = 7$ rings. In fact, it is possible to convert the honeycomb structure into the sigma approximant via a sequence of Stone–Wales transformations: A single transformation first generates $n = 5$ and 7 rings (Fig. 4a). Reapplying the concept of Stone–Wales transformation to the $n = 5$ and 6 rings, yields $n = 4$ and 7 rings (Fig. 4a). Thus, the stabilization of the $n = 7$ rings is characteristic to the full conversion of an initial honeycomb lattice into an ordered structure consisting of $n = 4$ and 7 rings only, according to:

$$12 \times Ti_6O_6 \rightarrow 4 \times Ti_5O_5 + 4 \times Ti_6O_6 + 4 \times Ti_7O_7 \ (1. \text{ Stone} - \text{Wales transformation})$$
$$\rightarrow 4 \times Ti_4O_4 + 8 \times Ti_7O_7 \ (2. \text{ Stone} - \text{Wales transformation})$$

This process converts 2/3 of all hexagonal rings to those with $n = 7$.

The unit cell areas of the honeycomb and the sigma phase are similar while their composition $Ti_{24}O_{36}$ is identical. By decorating 2/3 of the honeycomb $Ti_6O_6$ rings and all of the sigma-phase $Ti_7O_7$ rings with Ba as shown in Fig. 4b, c, structures of identical composition $Ba_8Ti_{24}O_{36}$ are derived. As detailed in the Supplementary Notes, DFT calculations were used to optimize the strains and atomic positions of each structure on a Pt substrate. In absence of decorating atoms, the honeycomb structure is favored, as expected. By contrast, Ba decoration leads to an energy lowering of 0.034 eV per $Ti_2O_3$ unit in favor of the sigma phase. This provides evidence that large A-type atoms drive Stone–Wales-type transformations to form $n = 7$ rings. This energy lowering is directly related to the reduction of the electrostatic dipole-dipole energy, which can be understood by the larger separation of the positively charged large A-type cation within the sigma phase: DFT shows that Ba atoms in the sigma phase have five closest Ba neighbors at an average distance of 6.96 Å, which is about 17% larger than for the honeycomb structure, 5.89 Å (this work) or 5.95 Å[29]. Also, the average height of the Ba atoms above the metal surface is 3.18 Å for the sigma structure, which corresponds to a reduction by 0.24 Å as compared to the honeycomb structure. Both, the larger separation and the smaller dipole moment, reduce the electrostatic dipole-dipole repulsion between the Ba ions and, therefore, support the sigma structure. On the other hand, rings with $n > 7$ are energetically unfavorable as in this case the A-O bond becomes too large. Finally, we suggest that the energy penalty of the $n = 4$ ring formation is overcompensated by the reduction of the electrostatic repulsion. Consequently, the nature of the long-range order should depend on the A-type ion coverage relative to the number of rings.

A coverage of 2/3 Ba per ring is obtained for the sigma phase, also known as the $3^2 4.3.4$ Archimedean tiling[30]. This QC approximant has been observed experimentally for Ba-Ti-O on Pt(111), Ru(0001), and Pd(111) surfaces[4,13,15]. In principle, a higher fraction of $n = 7$ rings and thus higher A-type atom density could be obtained if Stone–Wales transformations to form $n = 3$ rings were allowed, but we suspect that the energy penalty of $n = 3$ rings is too large. Instead, higher A-type atom density is obtained via the $n = 10$ rings associated with the short diagonal of the rhombus tile. By hosting two A-type atoms at a relatively short distance, they allow for a higher coverage than would be possible via Stone–Wales transformations alone. Two nearby oxygen atoms, which are not interconnected via additional Ti reduce the Coulomb repulsion between the A-type atoms. While there is not a Stone–Wales type transformation for generating an $n = 10$ ring, one can be formed by expanding the shared edge of a pair of adjoined $n = 6$ rings, adding another interior O, and moving the two interior O to be near the appropriate Ti atoms (Fig. 4d). The additional O atoms change the overall Ti:O ratio from 2:3. We hypothesize that the $n = 10$ ring of Fig. 4d lowers the (free) energy under relatively A-rich conditions,

allowing for triangle-square-rhombus tilings. Experimentally, such a transformation from a hexagonal honeycomb structure to the 48:18:6 quasicrystal approximant has been followed by LEED (Supplementary Fig. 1b, c).

For a Ba-decorated $Ti_2O_3$ honeycomb on Au(111), the honeycomb lattice is not modified up to 873 K and a coverage of 0.63[31]. In case of Ba-Ti-O on Pt(111), annealing at above $T = 1100$ K produces a variety of ordered structures[8,32] that have been interpreted as networks with $n = 7$ rings occupied by Ba[3]. For a coverage of 50% of all rings by an A-type atom, structures made from $n = 5$ and 7 rings only have been reported[3,32]. For coverages between 50% and 67%, a transition to a square-triangle tiling occurs. The sigma phase (Fig. 4) represents the highest Ba coverage in $n = 7$ rings that is realized in a pure square-triangle tiling. This is even by 5.1% higher than in a quasicrystalline square-triangle tiling[33] using the decoration scheme presented above. Thus, higher Ba coverages require the formation of $n = 10$ rings, which hosts two Ba atoms. Together with two neighboring $n = 7$ rings, the two A-type atoms form the vertices of a 30° rhombus, which is essential as the third tiling element for the formation of the dodecagonal structure of oxide QCs.

In summary, we have used a combination of SXRD experiments and DFT calculations to analyze the structure of an ultrathin film Sr-Ti-O approximant, which allows the structural solution of the directly related dodecagonal oxide quasicrystal. The structures of the quasicrystal and of the related quasicrystal approximants are based on 2D networks of interconnected $Ti_nO_n$ rings with $n = 4, 7$, and 10. We have shown that the formation of the oxide quasicrystal is governed by a fundamental principle which is based on the low-energy phase transformation (Stone–Wales transformation) in which the reduction of the electrostatic interaction between positively charged large cations located within the rings is the major driving mechanism. By exploiting this mechanism of ring size modifications, we expect that 2D structures of similar flexibility, as, e.g., made from $V_xO_y$, $FeWO_3$, $Nb_2O_3$, and $SiO_4$ units[20,34–41], might also be driven into oxide quasicrystals.

## Methods

### Experimental setup

Sample growth, LEED, and STM characterization were carried out in a UHV system operating at a base pressure of $1 \times 10^{-8}$ Pa. For thin film deposition, a four-pocket e-beam evaporator (EBE4, SPECS, Berlin) was used (certain commercial equipment and software are identified in this paper to adequately describe the methodology used. Such identification does not imply recommendation or endorsement by the National Institute of Standards and Technology, nor does it imply that the equipment and software identified is necessarily the best available for the purpose). Sr-Ti-O has been sublimed by heating a $SrTiO_3$ single crystal clamped between Ta plates. Additional evaporation of Ti can be supplied from a Ti rod. Evaporation rates were determined using a quartz crystal microbalance. SXRD measurements have been performed at the SixS beamline at the synchrotron SOLEIL. The UHV endstation of this beamline was used in which the diffractometer is coupled to standard UHV tools for sample preparation and analysis (LEED, STM). For temperature measurements, a pyrometer (Pyrospot DG40N, DIAS, $\lambda = 1600$ nm) was used at an emissivity of 0.17.

### Sample preparation

Ultrathin films of Sr-Ti-O have been grown onto Pt(111) at room temperature using molecular beam epitaxy. 8 Å of $SrTiO_3$ were deposited at an oxygen background pressure of $10^{-4}$ Pa with a rate of 1.0 Å/min. To balance Ti deficiencies in the film, 3 Å of Ti was added under the same conditions. The ultrathin films were post-annealed for 10 min at 950 K in $10^{-4}$ Pa of oxygen. Annealing for 20 min at 1150 K in UHV resulted in the formation of the 48:18:6 approximant homogeneously covering the Pt(111) single crystal. Such prepared samples were transferred through air to the SixS beamline. At the beamline, the samples

have been annealed at 850 K in $10^{-4}$ Pa oxygen atmosphere to remove residual carbon impurities originating from the air transfer. The approximant structure was recovered by UHV annealing at the beamline. Several annealing steps were performed to optimize the structural perfection as monitored by LEED prior to the SXRD characterization.

## SXRD analysis

The SXRD measurements were carried out at the UHV diffractometer of the SIXS beamline at Synchrotron SOLEIL in Paris, France. Monochromatic x-rays with photon energy of 11 keV were used to avoid Pt fluorescence (at 11.1 and 13 keV) and to reduce the background signal. The diffraction experiment was performed under grazing incidence at an angle of 0.2°. A 2D hybrid pixel XPad detector with $560 \times 240$ pixel was used for data collection. For the 48:18:6 approximant, the reciprocal space has been mapped in continuous rotation mode covering 120° of the azimuthal angles for polar angles ranging from 5 to 52° in increments of 0.1 and 1.5°, respectively. In case of the oxide QC, the reciprocal space map covered an azimuth of 182° at a polar angle variation from 4 to 40° in increments of 0.07 and 2°, respectively. The integration time was 0.3 s per frame. The 2D raw data were processed using the binoculars software[42] to generate 3D reciprocal voxel map data. For structure optimization and calculations of the unweighted residuum, the software package SHELXL was used[22].

## DFT calculations

DFT calculations were performed using the commercial software VASP (certain commercial equipment and software are identified in this paper to adequately describe the methodology used. Such identification does not imply recommendation or endorsement by the National Institute of Standards and Technology, nor does it imply that the equipment and software identified is necessarily the best available for the purpose)[43,44]. Projector-augmented-wave pseudopotentials[45,46] from the VASP library were used for each element. A Hubbard U correction was applied to the d-electrons of Ti[47,48]. Non-local dispersions were included[49,50]. Further details are given in the Supplementary Notes.

## Data availability

Source data are provided with this paper. The SXRD data generated in this study have been deposited in the Zenodo repository under the accession code https://doi.org/10.5281/zenodo.6787572. The DFT structure files and sample input files have been deposited in the Zenodo repository under the accession code https://doi.org/10.5281/zenodo.7007289.

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

## Acknowledgements

We are grateful for the attribution of beam time on the SIXS beamline of the Soleil Synchrotron. We acknowledge excellent support from the beamline stuff at the SIXS beamline and K. Mohseni. We like to thank R. Kulla for technical support. This work was funded in part by the Deutsche Forschungsgemeinschaft (DFG, German Research Foundation) – 406658237 and SFB 762.

## Author contributions

O.K. and S.S. prepared the samples. S.S., S.F., M.d.B., and W.W. performed the SXRD experiments. O.K. measured and analyzed STM and LEED data. S.S. and H.L.M. were involved in the SXRD analysis and structure solution. S.S., S.F., and W.W. developed decoration scheme and inflation rule. E.C. carried out the DFT calculations. S.F., E.C., H.L.M. and W.W. discussed the results and wrote the manuscript.

## Funding

## Competing interests

The authors declare no competing interests.
