## [Peer Review File · Nature Communications]

2D honeycomb transformation into dodecagonal quasicrystals driven by electrostatic forcesREVIEWER COMMENTS

Reviewer #1 (Remarks to the Author):

The paper brings crucial clarifications regarding the oxide quasicrystal layer structure. In particular, the new SXR diffraction data fit clearly establishes the Cockayne et al (2016) model as the only plausible solution.

The structure model itself receives further DFT-backed contextual justification by interpreting structural evolution of the (Ba/Sr)TiO layer from honeycomb and "sigma" states toward the quasicrystal phase, via Stone-Wales (like) transformations stabilized by Ba/Sr addition.

The paper is well written and sound in both methodology and conclusions. I recommend publication, after authors handle my notes regarding the Fig.1 (see further below). I also suggest authors to consider several comments/questions I had.

I would say, the square and triangle pieces of the decoration are now satisfactorily established, but I feel there are still open questions regarding the quasicrystal state:

1. the 30-degree rhombus: we still miss a strong energetic argument like the one presented for the square-triangle sigma state vs honeycomb, as it introduces isolated short Sr-Sr bonds.

2. The three pieces of the presented SrTiO decoration (triangle, square, rhombus) seem to warrant turning any (random) tiling of these tiles into valid oxide layer. The quasiperiodic NMG tiling is just one particular possibility. Looking at the Fig.1(a,d) and also extended data Fig.2a, I can see that Dodecagonal supertiles (D) with radius equal to the long diagonal of 30deg rhombus seem to entirely cover the plane, and they are highly correlated with the occurrence of the 30-deg rhombuses. Isn't that worth mentioning?

3. Along the lines of (2), I find structure in the STM of extended data Fig.2a particularly curious. As the caption states, I can see the horizontal and vertical rows indicating average periodicity. At the same time, looking at the pattern of abundant D supertiles, I cannot find clear periodic repetition from cell to cell. How does the periodicity emerge, when it is apparently not enforced by a tile-tile arrangement preferences?

4. Dodecagonal quasicrystal states are possible even without the rhombus tiles, but such OQC was not observed, to my knowledge. Then, is there any obvious reason why the occurrence of the rhombus tile seems to be prerequisite (not sufficient by itself of course) for OQC formation?

Finally, I have a comment regarding organization and caption content of the Fig.1:

- the order of introducing panels (a, d, b, e, c, f) is very confusing.
 - what do you mean by "tiling elements"? Elementary tiles?
 - from the caption, (d) is BaTiO QC state. Also, (e) is the QC state, distinguished by "circular arrangement of pairs of rhombuses". (remove "pairs"!)
- This is very confusing, since I cannot see any circularly arranged rhombuses in BaTiO QC state (d) (while (a) does seem to correspond with (b)). In (d), I only see zig-zag chains of rhombuses, and at least one isolated rhombus.

Reviewer #2 (Remarks to the Author):

Authors deal with the structure of dodecagonal oxide quasicrystals, originally found by the authors appeared in the Nature paper (2013). However, their structures have been controversial among researchers even in the authors' papers: say, STM and AFM studies cannot determine structures consisting of three elements with atomic resolution. Without doubt, it is important to clarify which

alkaline earth (Ba or Sr) or Ti atoms occupy the vertex positions of quasicrystal tiling model to understand the oxide quasicrystal. Here, they determine a large-scale approximant structure using SXRD and they confirm a previous theoretical model [3]. I thank corrections for [4][5]. As for DFT calculations, the refinement has been done in the paper as well. The whole collaboration work is sound, and data provided are well-prepared. However, the structural study for the same large approximant has been published [15], quite recently though. Moreover, the decoration of atoms in the Niizeki-Mitani-Gaehler dodecagonal tiling has been provided several years ago by the one of the authors [3]. In these respects, the novelty of the paper is not so much clear. Therefore, I cannot recommend its publication as it is.

I presume that the highlight of paper is the discussion part of the origin of the oxide quasicrystals. The authors tried to explain why oxide quasicrystals and approximants form. What is new is the introduction of the Stone-Wales transformations, which explains six-fold honeycombs transform to 5- and 7-membered rings; I like this generic argument. They explained the formation of sigma (33434) structures explicitly in terms of the Stone-Wales transformations, in which the driving force is the electrostatic repulsion. In my opinion, it is worth mentioning that the formation of squares by repulsion is a common feature to two-length scale potential models in soft-matter dodecagonal quasicrystals, while in the oxide system, Extended Figure 4 is quite important to explain the formation of rhombi or ten-membered rings, since this is a quite big difference between oxide quasicrystals and soft-matter dodecagonal quasicrystals without rhombus tiling. Thus Extended Figure 4 may be combined with Figure 4.

I would like to ask the authors that the importance and significance of the paper should be clearly presented at least in the abstract. The reason is the following. The KEY sentence in the abstract "Here we demonstrate the solution of the OQC structure by synchrotron-radiation based surface x-ray diffraction (SXRD) refinement of one of its approximants, which contains more than 300 atoms in the unit cell" is nothing to do with the title of the paper "From simple six-fold honeycombs to complex oxide quasicrystals: Self organization and long range order driven by electrostatic forces". I felt that the argument of the Stone-Wales transformations and the structure determination of the large scale approximant is not logically straight forward.

L.133-146: The scheme of half recursion rules of NMG tiling seems to be "first". Please clearly describe what is new. The paragraph is confusing between recursion and decoration to me.

Small corrections.

- L.125: Ref. 15 is Merchan et al.
- Figure 3 caption: Niizeki-Mitani-Gaehler or NMG for consistency?
- Extended Data Figure 2 caption L.2: 48:18:9 should be 48.18.6.

Reviewer #3 (Remarks to the Author):

The manuscript entitled "From simple six-fold honeycombs to complex oxide quasicrystals: Self organization and long range order driven by electrostatic forces" authored by Schenk, et al. provides a detailed analysis of real atomic structure of oxide quasicrystal, as well as their periodic approximants and simple honeycomb systems with a unified way. Oxide quasicrystals are known to form 2D systems with dodecagonal symmetry. Their atomic structure was tried to be modelled with triangle-and-square-based tilings, developed in the 1980s. However, the true tiling decoration for oxide quasicrystals and, especially, their approximants were not confirmed either experimentally nor by calculations up to date.

In the manuscript, the authors present a tiling model applicable for simple honeycomb structures to periodic approximants of oxide quasicrystals developed based on XRD structure analysis supported by DFT calculations (for large approximants). From the analysis a hierarchy of tiling models arises driven by electrostatic forces as introduced by Stone-Wales transformations, known in surface science developed for many 2D honeycomb lattices. For Sr/Ba-Ti-O systems, the authors show that atomic long-range order is guaranteed by titanium oxide rings Ti_nO_n formed as a result of the self-organization of Sr/Ba atoms. For small approximants they observed $n=4$ and

$n=7$ rings, whereas for large approximants additional $n=10$ rings occur with 2 Sr/Ba atoms inside, leading to the formation of a rhomb in the Niizeki-Gahler tiling. The authors provide a careful explanation of the formation of Ti-O rings supported by Stone-Wales transformations and confirm the stability of such systems by DFT calculations. This is the first detailed explanation of the formation of tiling in the approximants of oxide quasicrystals in the literature. Moreover, the authors introduce the novel inflation rule for 3 elements of Niizeki-Gahler tiling.

The organization of the material in the manuscript is good, the text is written clearly, and is easy to follow. The editorial side is good, the figures are presented in a good fashion. Conclusions are supported by a enough deep analysis. References are adequately cited and cover most recent results on the subject. The material presented in the manuscript is, in my opinion, highly interested in the community of material science, especially complex systems, providing insight into the long-range formation of atoms and the rules determining such formation. It is a vital question for all aperiodic materials that still awaits an answer. 2D oxide quasicrystals and their approximants are still a rather new topic in the field but are gaining interest from experimentalists and theoreticians. Understanding the formation of oxide quasicrystals is a step toward a better understanding of the nature of the quasicrystals. The results presented are well justified by experimental observations and theoretical analyses, including total-energy calculations. I think it is worth publishing in Nature Communications, as it will find interest among a broad range of scientists.

In the following, I give some more specific comments and questions.

- 1) From previous literature reports (e.g., PRL 117, 095501 (2016) or Phys. Rev. Mat. 4, 103402 (2020)) it is known that squares and triangles of whatever tiling it is (NMG or Stampfli-Gahler or DFT-base Cockayne, et al., model) are distorted. It appears to also be necessary in this case, as seen in Figure 2 (a or b). The authors could discuss the scale of distortion as compared to the ideal tiling model (it seems to be different for a given tile in different locations). What is the nature of this distortion? Is it purely a structural effect (like forcing by chemical concentration and then confirmed by energy minimization) or a signature of some defects (like phonons or phasons). The authors in a small section could discuss the distortion of tiles.
- 2) In previous papers of the same group (e.g., Schenk et al. 2017 J. Phys.: Condens. Matter 29 134002) dealing also with approximant of Sr-Ti-O a Stampfli-Gahler tiling is considered as the most promising tiling model for this system. There are also a quite large number of Stampfli-like tilings reported in the literature. The authors could explain in short the difference in the NMG and Stampfli-Gahler triangle-square-rhomb models and discuss the superiority of one over another.
- 3) The authors should give a chemical formula for the two approximants (relative concentration of Sr/Ti/O). Subsequently, the large approximant (48:18:6) apparently contains more oxygen. I do not clearly see whether the Sr/Ti ratio changes? What would be expected for a quasicrystal? What should be the (estimated) ratio of building blocks and their decoration? As suggested, e.g., in Yuhara et al., Phys. Rev. Mat. 4, 103402 (2020) transition from periodic to aperiodic system should be accompanied by a decrease of the Ti / Ba ratio. How would a transition to quasicrystal in your case affect the structure model and tiling decoration? If the authors have a background to this division, I would expect to see it in the manuscript.
- 4) Just a small editorial remark: In Figure 2a I would not call the arrows and tiles of magenta color, but rather of a purple color.

RESPONSE TO REVIEWERS' COMMENTS

We thank all reviewers for their fast response.

Reviewer #1 judges *“The paper brings crucial clarifications regarding the oxide quasicrystal layer structure.”* He recommends publication after consideration of the following comments:

1) The 30° rhombus: we still miss a strong energetic argument like the one for the square-triangle sigma state vs the honeycomb, as it introduces isolated short Sr-Sr bonds.

It is true that the 30° rhombus shortens the Sr-Sr distance and thus increases the *unscreened* Sr-Sr dipole repulsion. At the same time (see revised Figure 4(d)), the screening is increased because there is now a pair of oxygen atoms sandwiched between the two Sr instead of a single oxygen. Furthermore, the increased Sr-Ti distances decrease electrostatic repulsion between Sr and Ti. We hypothesize that net effect is to reduce the overall (free) energy under the conditions of OQC formation. We would like to be able to demonstrate this via ab initio calculations as we did for the honeycomb vs sigma example, but the introduction of 30° rhombi to the tiling decoration scheme changes the composition and precludes a before and after comparison unless chemical potentials are introduced. We emphasize that the new shorter Sr-Sr distance created allows the density of the Sr ions to be increased beyond that of square-triangle tilings. As long as the (free) energy change induced by the rhombus is favorable enough, we expect tilings of squares, triangles and rhombi to form in conditions where the equilibrium Sr concentration is higher than in the sigma phase. We already make the connection between higher Sr density and rhombi in the manuscript, but have added an explicit physical motivation to the text:

“We hypothesize that the $n=10$ ring of Fig. 4d lowers the (free) energy under relatively A-rich conditions, allowing for triangle-square-rhombus tilings.”

2) The three pieces of the presented SrTiO decoration (triangle, square, rhombus) seem to warrant turning any (random) tiling of these tiles into valid oxide layer. The quasiperiodic NMG tiling is just one particular possibility. Looking at the Fig.1(a,d) and also extended data Fig.2a, I can see that Dodecagonal supertiles (D) with radius equal to the long diagonal of 30deg rhombus seem to entirely cover the plane, and they are highly correlated with the occurrence of the 30-deg rhombuses. Isn't that worth mentioning?

Indeed this supertile is an important feature of the Niizeki-Gähler tiling (NGT – we use consequently now this term instead of NMG to adopt to the existing literature), especially when it comes to the higher-hierarchical level of self-similarity in the quasicrystal structure. This supertile has been thoroughly discussed in previous publications (e.g. Schenk et al., Acta Cryst. A75, 307 (2019)) and is not relevant in the context of the work presented here. However, for completeness we refer to the supertile in the caption of Figure 1 and modified the main body by stating:

“The latter include chains of three rhombi in the approximant, while pairs of rhombi dominate the OQC tiling meeting in the centre of a characteristic dodecagon [11]“

3) Along the lines of (2), I find structure in the STM of extended data Fig.2a particularly curious. As the caption states, I can see the horizontal and vertical rows indicating average periodicity. At the same time, looking at the pattern of abundant D supertiles, I cannot find clear periodic repetition from cell to cell. How does the periodicity emerge, when it is apparently not enforced by a tile-tile arrangement preferences.

The outline of four unit cells are added to Extended Data Fig.2a for guidance. We are confident that this change allows a clear identification of the periodically repeating motifs despite of some naturally occurring defects.

4) Dodecagonal quasicrystal states are possible even *_without_* the rhombus tiles, but such OQC was not observed, to my knowledge. Then, is there any obvious reason why the occurrence of the rhombus tile seems to be prerequisite (not sufficient by itself of course) for OQC formation.

The dodecagonal triangle-square-rhombus tiling is the easy way for increasing the Ba/Sr density in the system significantly. By applying the decoration scheme presented in this work to the dodecagonal square-triangle tiling by Stampfli (P. Stampfli, Helvetica Physica Acta (1986) 59 1260-1263 – included as new citation 33), 5.1% less $n=7$ rings are available for the Ba decoration than in the sigma phase. Only by allowing Ba ions in shorter distance due to the formation of rhombs the density is increased by +10% upon the sigma phase value. We added additional explanations to this point to the main text by stating:

“The sigma phase (Fig. 4) represents the highest Ba coverage in $n=7$ rings that is realized in a pure square-triangle tiling. This is even by 5.1% higher than in a quasicrystalline square-triangle tiling [33] using the decoration scheme presented above. Thus...”

All subsequent citations have been update.

5) Comments regarding organization and caption content of the Fig.1:

- the order of introducing panels (a, d, b, e, c, f) is very confusing.
 - what do you mean by "tiling elements"? Elementary tiles?
 - from the caption, (d) is BaTiO QC state. Also, (e) is the QC state, distinguished by "circular arrangement of pairs of rhombuses". (remove "pairs"!)
- This is very confusing, since I cannot see any circularly arranged rhombuses in BaTiO QC state (d) (while (a) does seem to correspond with (b)). In (d), I only see zig-zag chains of rhombuses, and at least one isolated rhombus..

This figure has been reorganized and the caption modified following to the suggestions.

Reviewer #2 values the importance of our approach for deciding between the opposing structural models and provides positive feedback to the quality of our work in general and of the current manuscript. However, he has concerns about the novelty of our work in view of the recent publication on the 48:18:6 approximant.

We very much appreciate your decision to overrule the novelty concerns raised by reviewer #2. The previous work (Merchan et al. 2022) was the first to report the existence of the 48:16:6 approximant and its compatibility with the Cockayne et al. model. In the structure determination section of our paper, we use SXR structure refinement to definitively decide between *different* models, in itself a significant advance.

Reviewer #2 raised additional issues, which are addressed in the following paragraphs:

1)... Extended Figure 4 is quite important to explain the formation of rhombi or ten-membered rings, since this is a quite big difference between oxide quasicrystals and soft-

matter dodecagonal quasicrystals without rhombus tiling. Thus Extended Figure 4 may be combined with Figure 4.

We perfectly agree and combined both Figures. Thus, the relevance of the rhombus to the tiling and to quasicrystal formation is now more clear.

The formation of square tiles and square-triangle tiling quasicrystals in soft matter as a consequence of certain interaction potentials with two length scales is interesting. In the OQCs, there is no length scale for the dipolar interactions, instead, we argue that square-triangle and square-triangle-rhombus tilings arise from minimizing dipole-dipole repulsion under the constraint of a threefold network. The different tilings in the two cases arise because the underlying physics is different. An interesting topic for future work would be to devise tiling Hamiltonians for each case to quantify their energetic differences.

2) I would like to ask the authors that the importance and significance of the paper should be clearly presented at least in the abstract. The reason is the following. The KEY sentence in the abstract “Here we demonstrate the solution of the OQC structure by synchrotron-radiation based surface x-ray diffraction (SXRD) refinement of one of its approximants, which contains more than 300 atoms in the unit cell” is nothing to do with the title of the paper “From simple six-fold honeycombs to complex oxide quasicrystals: Self organization and long range order driven by electrostatic forces”. I felt that the argument of the Stone-Wales transformations and the structure determination of the large scale approximant is not logically straight forward.

The ability to decide upon the correct structural model for understanding OQCs lays the foundations for solving their formation mechanism. For this reasons both subjects are equally important for the manuscript. The title is fully justified from the last part of the abstract in which we state: “*DFT calculations demonstrate that the presence of large alkaline earth metal atoms and the reduction of their mutual electrostatic repulsion drives the $n=6$ structure of the 2D Ti_2O_3 honeycomb arrangement via Stone-Wales transformations into an ordered structure with empty $n=4$ and occupied $n=7$ rings.*”

L.133-146: The scheme of half recursion rules of NMG tiling seems to be “first”. Please clearly describe what is new. The paragraph is confusing between recursion and decoration to me.

In contrast to the previously published recursion rule we now present a solution that results in an overlap free NGT. We added considerations how to derive the stoichiometry of an approximant structure from the decoration scheme to this paragraph, which hopefully helps to sort out the differences between decoration and recursion.

Small corrections.

- L.125: Ref. 15 is Merchan et al. Corrected.
- Figure 3 caption: Niizeki-Mitani-Gaehler or NMG for consistency?

For consistency we now make use of Niizeki-Gähler tiling in the whole manuscript.

- Extended Data Figure 2 caption L.2: 48:18:9 should be 48.18.6. Corrected.

Reviewer #3 provides a positive feedback to the paper as a whole. He thinks “... *it is worth publishing in Nature Communications, as it will find interest among a broad range of scientists*” We address his specific comments and questions in the following:

1) From previous literature reports (e.g., PRL 117, 095501 (2016) or Phys. Rev. Mat. 4, 103402 (2020)) it is known that squares and triangles of whatever tiling it is (NMG or Stampfli-Gähler or DFT-base Cockayne, et al., model) are distorted. It appears to also be necessary in this case, as seen in Figure 2 (a or b). The authors could discuss the scale of distortion as compared to the ideal tiling model (it seems to be different for a given tile in different locations). What is the nature of this distortion? Is it purely a structural effect (like forcing by chemical concentration and then confirmed by energy minimization) or a signature of some defects (like phonons or phasons). The authors in a small section could discuss the distortion of tiles.

Indeed, the sigma-phase structure in Ba-Ti-O/Pt(111) (PRL 117, 095501 (2016) or Phys. Rev. Mat. 4, 103402 (2020)) exhibits significant distortions. In contrast, on a Ru(0001) surface the sigma-phase tiling appears much less distorted (Phys. Status Solidi B 257, 1900655 (2020)). For the OQC in Ba-Ti-O/Pt(111) the statistical analysis also proved an excellent homogeneity in the geometry of tiles (Acta Cryst. A75, 307 (2019)). In the present Sr-Ti-O/Pt(111) system, the ideal tiling nicely coincides with the positions of the Sr vertices in atomically-resolved STM images (Fig. 1a) and also the area integrating SXR (Fig. 2a) does not report severe deviations from the ideal positions. The DFT calculations are naturally more prone to local relaxations. But also in there, major deviations are only seen in the more flexible Ti_nO_n network. Hence we attribute the formerly observed changes to a particular case of substrate-registry induced distortions, which is why we refrain from discussions of this detail in the current manuscript.

2) In previous papers of the same group (e.g., Schenk et al. 2017 J. Phys.: Condens. Matter 29 134002) dealing also with approximant of Sr-Ti-O a Stampfli-Gähler tiling is considered as the most promising tiling model for this system. There are also a quite large number of Stampfli-like tilings reported in the literature. The authors could explain in short the difference in the NMG and Stampfli-Gähler triangle-square-rhomb models and discuss the superiority of one over another.

The comprehensive statistical analysis reported by Schenk et al. (Acta Cryst. A75, 307 (2019)) confirmed that the Niizeki-Gähler tiling provides the correct mathematical model of OQC systems. In the first publication reporting the discovery of OQCs (Nature 502, 215-218 (2013)) the term Stampfli-Gähler tiling was introduced by mistake and we are very sorry for causing confusion. Stampfli described a dodecagonal square-triangle tiling, while Gähler as well as Niizeki and Mitani described the square-triangle-rhomb tiling of OQCs. We are now perfectly aware of the differences. The term Niizeki-Gähler tiling is well-established nowadays and we adopted it also for the current manuscript.

3) The authors should give a chemical formula for the two approximants (relative concentration of Sr/Ti/O). Subsequently, the large approximant (48:18:6) apparently contains more oxygen. I do not clearly see whether the Sr/Ti ratio changes? What would be expected for a quasicrystal? What should be the (estimated) ratio of building blocks and their

decoration? As suggested, e.g., in Yuhara et al., Phys. Rev. Mat. 4, 103402 (2020) transition from periodic to aperiodic system should be accompanied by a decrease of the Ti / Ba ratio. How would a transition to quasicrystal in your case affect the structure model and tiling decoration? If the authors have a background to this division, I would expect to see it in the manuscript.

Now that the Cockayne model has been confirmed, exact stoichiometries of all possible structures can be derived from the decoration scheme. It exactly tells in which ratio Ba/Sr, Ti, and O are mixed, when a structure contains a given amount of triangles, squares and rhombs. For the 48:18:6 approximant the composition $\text{Sr}_{48}\text{Ti}_{132}\text{O}_{204}$ and stoichiometric ratio of 0.36:1:1.55 follows. The ratio of the OQC is 0.37:1:1.55.

We now state in the text: "For the 48:18:6 approximant a composition of $\text{Sr}_{48}\text{Ti}_{132}\text{O}_{208}$ and Sr:Ti:O ratio of approximately 0.36:1:1.55 is obtained. For the OQC, the Sr:Ti:O ratio is very similar: $(\sqrt{3} - 1)/2:1:(3\sqrt{3} + 1)/4 \approx 0.37:1:1.55$ ".

4) Just a small editorial remark: In Figure 2a I would not call the arrows and tiles of magenta color, but rather of a purple color.

Has been corrected to purple.

We again thank the reviewers for their comments, which help to improve the manuscript in various aspects. We furthermore made some changes to the manuscript to meet the requirements of Nature Communications: The title has been shortened to 15 words, the abstract has been reduced to a length below 150 words, the corresponding author has been labeled, and a separate Data Availability section has been included.

With the applied changes we are looking forward to publication in Nature Communications.

Sincerely,
Dr. Stefan Förster

Martin-Luther-Universität Halle-Wittenberg
Von-Danckelmann-Platz 3, 06120 Halle (Saale), Germany

REVIEWERS' COMMENTS

Reviewer #1 (Remarks to the Author):

I consider the current revision as satisfactory and recommend publication of the article.

Reviewer #2 (Remarks to the Author):

Now I think the paper merits Nature communications after reconsidering the following.

- Still I am not satisfied with the revised abstract.
 - An unreferenced abstract is required. (Editorial)
 - The word (the role) of “DFT” is totally missing in contrast to the previous one. I don't think it's a good decision.

- L132, L133:

IF $\text{Sr}_{48}\text{Ti}_{132}\text{O}_{208}$ was correct, the calculated ratio 0.36:1:1.55 would not be correct, since

$$\frac{48}{132} = 0.3636\cdots, \frac{208}{132} = 1.5757\cdots, \frac{\sqrt{3}-1}{2} = 0.3660\cdots, \frac{3\sqrt{3}+1}{4} = 1.5490\cdots$$

However, $\text{Sr}_{48}\text{Ti}_{132}\text{O}_{204}$ is correct, the calculated ratio seems to be correct, since

$$\frac{204}{132} = 1.5454\cdots, \frac{3}{2} \times 48 + 6 \times 18 + 4 \times 6 = 204$$

In my opinion, three decimal places are better to compare the compositions.

Reviewer #3 (Remarks to the Author):

My questions were all addressed and issues were clarified. I recommend publishing the manuscript with no further changes.

Response to Reviewer #2

1) An unreferenced abstract is required. (Editorial) The word (the role) of “DFT” is totally missing in contrast to the previous one.

We removed the references from the abstract to comply with the editorial standards of Nature Communications. The role of DFT has been included in the abstract. The sequence of references has been modified accordingly.

2) The reviewer noticed a mistake in the given stoichiometric ratio.

We are grateful for the careful review of the manuscript. Indeed the stoichiometry of the compound must be $\text{Sr}_{48}\text{Ti}_{132}\text{O}_{204}$. We corrected our mistake in the manuscript. As mentioned by the reviewer, the calculated ratios were correctly given. According to his/her suggestions we modified the ratios to display one more digit.

Please accept our following suggestion for a brief summary of the main findings of the paper to be used on your website:

“Quasicrystals are perfectly ordered crystals lacking translational symmetry. Here the authors unravel the formation mechanism of two-dimensional dodecagonal quasicrystals that arise from systematic modifications of a hexagonal honeycomb structure.”

We thank for being accepted for publication in Nature Communication and hope to provide you here with all necessary details to proceed in the production process without further delay.

Sincerely,
Dr. Stefan Förster

Martin-Luther-Universität Halle-Wittenberg
Von-Danckelmann-Platz 3, 06120 Halle (Saale), Germany